# In silico-labeled ghost cytometry

**Masashi Ugawa[1,2,3†], Yoko Kawamura[1†], Keisuke Toda[1], Kazuki Teranishi[1], Hikari Morita[1], Hiroaki Adachi[1], Ryo Tamoto[1], Hiroko Nomaru[1], Keiji Nakagawa[1], Keiki Sugimoto[1], Evgeniia Borisova[4], Yuri An[4], Yusuke Konishi[5], Seiichiro Tabata[5], Soji Morishita[6], Misa Imai[6], Tomoiku Takaku[6], Marito Araki[6], Norio Komatsu[6], Yohei Hayashi[4], Issei Sato[1,3], Ryoichi Horisaki[1,3,7], Hiroyuki Noji[3], Sadao Ota[1,3,7]***

[1]Thinkcyte Inc, Tokyo, Japan; [2]Center for Advanced Intelligence Project, RIKEN, Tokyo, Japan; [3]The University of Tokyo, Tokyo, Japan; [4]BioResource Research Center, RIKEN, Tsukuba, Japan; [5]Sysmex Corporation, Kobe, Japan; [6]Juntendo University, Tokyo, Japan; [7]PRESTO, Japan Science and Technology Agency, Kawaguchi, Japan

**Abstract** Characterization and isolation of a large population of cells are indispensable procedures in biological sciences. Flow cytometry is one of the standards that offers a method to characterize and isolate cells at high throughput. When performing flow cytometry, cells are molecularly stained with fluorescent labels to adopt biomolecular specificity which is essential for characterizing cells. However, molecular staining is costly and its chemical toxicity can cause side effects to the cells which becomes a critical issue when the cells are used downstream as medical products or for further analysis. Here, we introduce a high-throughput stain-free flow cytometry called in silico-labeled ghost cytometry which characterizes and sorts cells using machine-predicted labels. Instead of detecting molecular stains, we use machine learning to derive the molecular labels from compressive data obtained with diffractive and scattering imaging methods. By directly using the compressive 'imaging' data, our system can accurately assign the designated label to each cell in real time and perform sorting based on this judgment. With this method, we were able to distinguish different cell states, cell types derived from human induced pluripotent stem (iPS) cells, and subtypes of peripheral white blood cells using only stain-free modalities. Our method will find applications in cell manufacturing for regenerative medicine as well as in cell-based medical diagnostic assays in which fluorescence labeling of the cells is undesirable.

***For correspondence:**
sadaota@solab.rcast.u-tokyo.
ac.jp

[†]These authors contributed equally to this work

## Editor's evaluation

This paper explores a novel approve to sorting cells without the use of fluorescent labeling using a light diffraction method called ghost cytometry. This paper first demonstrates this capability with commercial cell lines and then sorting hematopoietic cells from a patient sample.

## Introduction

Characterization and isolation of a large number of single live cells are of critical importance in life sciences and medicine (*Dean et al., 2005*; *Dudley et al., 2002*; *Fraietta et al., 2018*; *Knoepfler, 2009*; *Reya et al., 2001*). During qualitative or quantitative characterization of cells, it is most popular to use molecular staining such as those adopting antibody-antigen binding, where fluorescent markers are physically added (*Brown and Wittwer, 2000*). Combined with a magnetic or flow-based sorting system, this enables selective isolation of the cells of interest at high throughput (*Herzenberg et al., 1976*; *Miltenyi et al., 1990*; *Shapiro, 2005*). However, molecular staining has numerous disadvantages: the staining process is costly in terms of both time and money; number of detectable markers with fluorescence detection are limited by the spectral overlap of fluorophores (*Perfetto et al., 2004*);

multiple spectral compensation is complex and time consuming (*Roederer, 2001*); immunocytochemistry staining is inconsistent due to the wide variability of antibodies (*Burry, 2011*; *Weigert et al., 1970*); and molecular staining potentially causes side effects to the cells by chemical toxicity (*Fried et al., 1982*; *Patil et al., 2018*; *Progatzky et al., 2013*). These disadvantages become an issue in medical applications. For instance, when producing cell therapy products, the chemical toxicity of molecular staining may affect the final product. Another example is that, during medical diagnosis, the labor and financial cost limits the tests to be used in only large hospitals that can afford them.

On the other hand, microscopic image-based cell classification of unstained cells is free from such limitations of molecular labeling and is a promising approach for evaluating cell functions or potentials in fields such as cell manufacturing (*Buggenthin et al., 2017*; *Chang et al., 2017*; *Niioka et al., 2018*). Although unstained cell images were thought to lack specific biomolecular information, recent studies (*Christiansen et al., 2018*; *Ounkomol et al., 2018*) revealed the potentiality of bridging between images and molecular labels: prediction of labels from imaging information or 'in silico labeling.' However, these approaches utilizing conventional microscopes sacrifice the high throughput and sorting capability which exist in molecular labeling methods (*Han et al., 2016*; *Lindström, 2012*). This restriction arises from the fact that the acquisition of microscopy images is slow, and the processing of high-content images is even slower (*Pepperkok and Ellenberg, 2006*). Especially, to achieve real-time image-based cell analysis and isolation at high throughput, the processing speed becomes one of the substantial bottlenecks (*Han et al., 2016*; *Nitta et al., 2018*). In contrast, conventional flow-based cell sorting systems that process simple cell information such as total fluorescence intensity fast enough to operate at around 10,000 cells/s (*Sutermaster and Darling, 2019*).

To achieve in silico-labeling-based cell classification at high throughput, we hypothesized that we do not need to construct the two-dimensional (2D) or three-dimensional 'images' for computationally predicting the labels. Previously, we have demonstrated ghost cytometry, an 'imaging' cytometry that is able to predict cell types based on fluorescence signals encoded with morphological information of cells (*Ota et al., 2018*). However, the usage of fluorescence staining was inevitable and, therefore, could not suffice applications where cell staining is unfavorable. In this work, we developed a biologically supervised imaging cytometry method free from fluorescence labels called in silico-labeled ghost cytometry (iSGC) while we do not intend to produce any full 2D images in this work to prioritize the speed, we use terms of image or imaging for simplicity. This method performs ultrafast real-time prediction of biological labels by machine learning-based analysis of compressively measured image information of cells in a processing time of down to the order of microseconds on a field-programmable gate array (FPGA). Herein, the machine learning models are pre-trained by the stain-free imaging data and biological labels obtained by fluorescence staining; afterward, it becomes able to predict the biological labels without the measurement of fluorescence labeling. In other words, iSGC performs high-throughput flow cytometry without fluorescence stains but as if cells are fluorescently stained.

## Results
### Principle of iSGC and cell sorting
In iSGC, morphological information of the cells is obtained with a stain-free compressive ghost imaging technique which we call diffractive ghost motion imaging (dGMI). Similar to previously reported fluorescence ghost cytometry (*Ota et al., 2018*), an imaging waveform is obtained from cells passing through a structured illumination at a speed where the signal width is shorter than 100 µs (*Figure 1—figure supplement 1E*). However, instead of collecting the whole photons with a bucket detector, to perform stain-free imaging, one to several fringe patterns in the diffractive speckle pattern from the transmitted light are acquired with a single-pixel detector (*Figure 1—figure supplement 1*). While full image reconstruction was demonstrated either with iterative acquisition using different structured illuminations or structured masks with a single-pixel detector (*Horisaki et al., 2017a*; *Horisaki et al., 2017b*), iSGC adopts a reduced acquisition for performing high-speed cell classification (*Figure 1—figure supplement 1* and *Figure 1—figure supplement 2*).

*Figure 1* depicts an overall workflow that enables biologically supervised in silico labeling of ultrafast dGMI signals without image production. We first prepare a training data set by simultaneously acquiring the dGMI waveform and the fluorescence label from each cell. We then train the machine

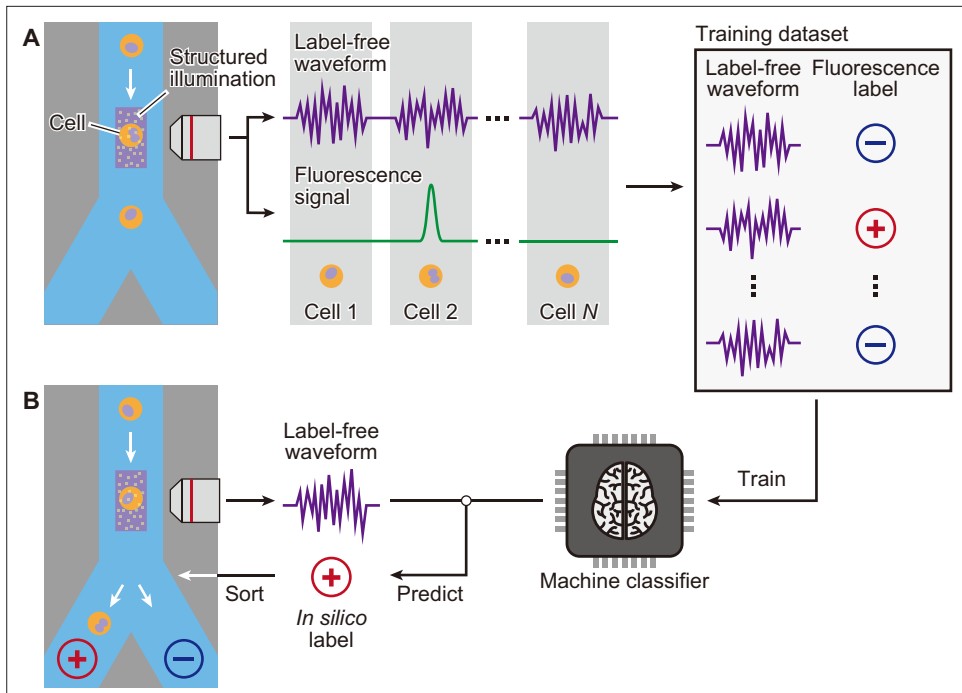

**Figure 1.** Schematic of iSGC to analyze and sort cells based on machine-predicted labels. (**A**) First, the training data set is prepared by simultaneously acquiring the stain-free waveform and the fluorescence signal from each cell. Using the fluorescence intensity obtained from the fluorescence signal, each cell is labeled as positive or negative according to a defined threshold of the fluorescence intensity. The machine classifier is trained using this data set comprised of stain-free waveforms and fluorescence labels. (**B**) Once trained, in turn, the classifier can predict the fluorescence label in silico from the stain-free waveforms and sort the cells upon this prediction in real time. iSGC, in silico-labeled ghost cytometry.

The online version of this article includes the following figure supplement(s) for figure 1:

**Figure supplement 1.** Principle of dGMI and ssGMI.

**Figure supplement 2.** Principle of BSC, bsGMI, and fsGMI.

**Figure supplement 3.** Optical system for iSGC.

classifier using this data set of biologically labeled waveforms. Once the classifier is trained, in turn, it is able to predict the fluorescence label from the stain-free imaging waveform. As a proof-of-concept demonstration, we here performed iSGC with two cell-line samples of HeLa S3 cells and MIA PaCa-2 cells (*Figure 2*). The two cell lines are similar in size (*Figure 2A*) and mixed at equal concentration with only one of them stained with green fluorescence using LIVE/DEAD Fixable Green Dead Cell Stain (Invitrogen) (*Figure 2B*). This mixture was hydrodynamically focused and flowed through the structured illumination within a microfluidic channel, where the dGMI waveform and green fluorescence of the cells were obtained simultaneously. Using the training data set comprised of the stain-free dGMI waveforms and the fluorescence labels, we trained a classifier based on support vector machine (SVM) (*Boser et al., 1992*). SVMs are stable in terms of optimization, and their generalization abilities are sufficiently analyzed on the basis of statistical learning theory. When the trained classifier was implemented on a FPGA, it judged the cells in real time using only the stain-free dGMI waveforms and then enabled subsequent sorting of the cells according to the judgment (*Figure 1B*). The actual time lapsed during the judgment for a single cell with the FPGA was 6.0 μs. In iSGC, using a high-content modality, the classifier can find a decision boundary in a higher-dimensional space and is able to make decisions based on the complex information. As a result, iSGC was able to infer the fluorescence label corresponding to the cell type in this first case from a fluorescence-free measurement at a high area under the receiver operating characteristic curve (AUC) of 0.963 for cells flowed after training (*Figure 2C and D*). In the actual sorting process, iSGC classified the cells at an accuracy of 0.917 according to the FPGA classifier (*Figure 2—figure supplement 1C*), and the purity of the

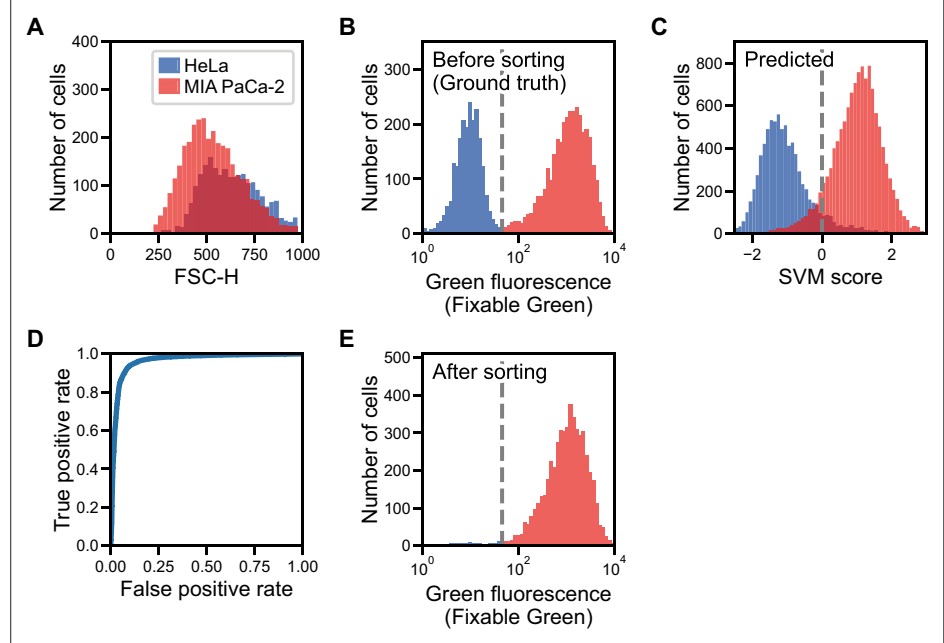

**Figure 2.** Label prediction and cell sorting with iSGC using cell line samples. (**A**) Histogram of FSC for HeLa S3 (blue) and MIA PaCa-2 (red) cells. The intensity of FSC reflects the size of cells. There is a large overlap of the two cell-line populations, showing that the two cell lines are similar in size. (**B**) Fluorescence intensity evaluation of HeLa S3 and MIA PaCa-2 cells before sorting. The population with lower green fluorescence intensity corresponds to HeLa S3 cells while the population with higher fluorescence intensity corresponds to MIA PaCa-2 cells. (**C**) SVM score histogram for the classification of HeLa S3 and MIA PaCa-2 cells obtained by iSGC as in silico-predicted labels during sorting. Blue and red colors assigned in the histogram correspond to the cells labeled as HeLa S3 and MIA PaCa-2 cells, respectively, derived from the green fluorescence obtained simultaneously for validation. The dashed line shows the decision boundary by the SVM. (**D**) A receiver operating characteristic curve for the classification of HeLa S3 and MIA PaCa-2 cells using dGMI during sorting. The AUC for the classification with dGMI was 0.963, showing a high classification ability. (**E**) Fluorescence intensity evaluation of HeLa S3 and MIA PaCa-2 cells after sorting. The concentration of MIA PaCa-2 cells was enriched from 60.3% (**B**) to 97.3% (**E**). dGMI, diffractive ghost motion imaging; FSC, forward scatter; iSGC, in silico-labeled ghost cytometry; SVM, support vector machine.

The online version of this article includes the following figure supplement(s) for figure 2:

**Figure supplement 1.** Comparison with FSC and SSC and confusion matrix for the sorting of HeLa S3 and MIAPaCa-2 cells with iSGC.

**Figure supplement 2.** Gating for the flow cytometry analysis in the classification and sorting of HeLa S3 cells and MIA PaCa-2 cells.

actual sorted cells was 97.3% (*Figure 2E*). In contrast, such accurate and high-speed classification and selection are challenging using existing flow cytometers that rely on low-content stain-free data such as forward scatter (FSC), side scatter (SSC), or back scatter (BSC), which is SSC's analogy and can be used interchangeably with SSC (*Figure 1—figure supplement 2*). Using only FSC and SSC, the SVM-based classifier can only draw a decision boundary in 2D space, and the classification had an AUC of only 0.936±0.005 (gray dashed line in *Figure 2—figure supplement 1B*).

## Classification of induced pluripotent stem cell (iPSC)-derived cells

iSGC exhibits its significant potential in cell manufacturing processes in applications including regenerative medicine and cell therapy, wherein the quality of cells has to be monitored and controlled from various aspects including viability, purity, and identity based on cell type and differentiation states (*Kolkundkar, 2014*; *Morgan et al., 2006*; *Segers and Lee, 2008*; *Yoshihara et al., 2017*). We here demonstrate the iSGC-based cell analysis in a cell production line using human iPSCs. The production line starts from thawing frozen-preserved iPSCs which are then passed through multiple differentiation steps, leading to the final cell product. Throughout this production line, monitoring viability,

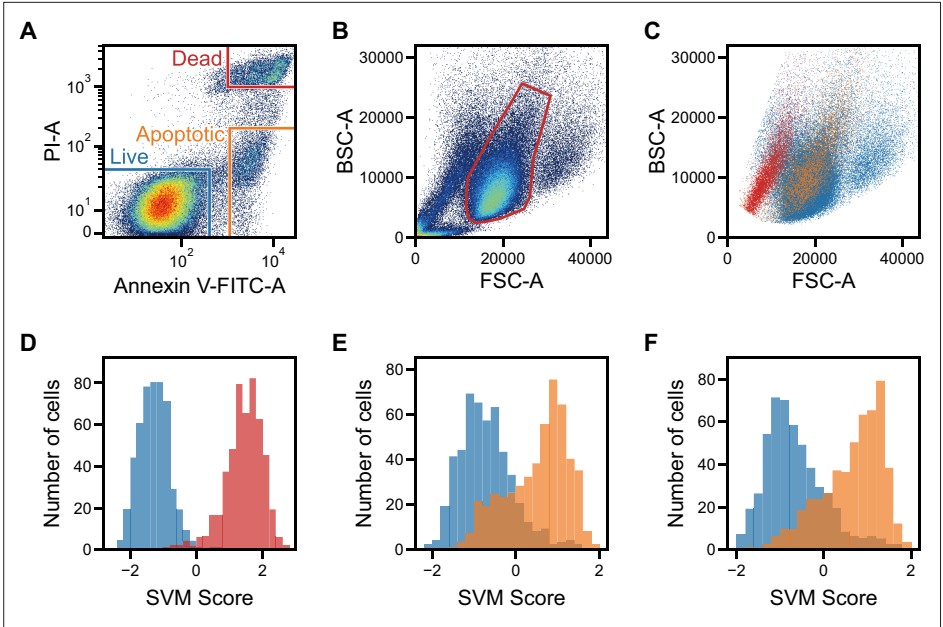

**Figure 3.** Classification of live, dead, and apoptotic iPSCs with iSGC. (**A**). Scatter plot of PI and Annexin V for iPSCs. The population within the blue, red, and orange regions were labeled as live, dead, and apoptotic cells, respectively. (**B**). Scatter plot of FSC and BSC for iPSCs without exclusion of debris and doublets. (**C**). Scatter plot of FSC and BSC for each labeled iPSC populations. The blue, red, and orange dots each correspond to live, dead, and apoptotic cells, respectively. All populations, which are previously shown in (**B**), are gated prior to labeling for removing debris and doublets (***Figure 3—figure supplement 1A*** and ***Figure 3—figure supplement 2***). In the plot, the live and dead populations have distinct separation, but the live and apoptotic populations overlap. (**D–F**) SVM score histograms for the iSGC-based classification of dead cells from live cells (**D**), apoptotic cells from live cells (**E**), and apoptotic cells from live cells within the red region in the scatter plot (**B, F**). The colors correspond to the labels in (**A**). All histograms are the best result of 10 times random sampling. The AUCs for each classification were 0.999 (**D**), 0.885 (**E**), and 0.904 (**F**). The mean and standard deviation of AUCs for the 10 trials in each condition were 0.998±0.002, 0.877±0.007, and 0.891±0.012, respectively (***Figure 3—figure supplement 1***, D, E, and F). BSC, back scatter; FSC, forward scatter; iPSC, induced pluripotent stem cell; iSGC, in silico-labeled ghost cytometry; PI, propidium iodide; SVM, support vector machine.

The online version of this article includes the following figure supplement(s) for figure 3:

**Figure supplement 1.** Scatter plot, confusion matrices, and SVM score histograms for the classification of live, dead, and apoptotic cells using iPSCs with iSGC.

**Figure supplement 2.** Gating for the iSGC analysis in the classification of live, dead, and apoptotic iPSCs.

liveliness, expressions states, and purity of the cell population, and often their selective enrichment are of critical importance. Up to now, these examinations have required molecular staining, which is often toxic to cells and, if not toxic, the cells can be affected, for instance, by immune response (***Progatzky et al., 2013***). Thereby, we introduce iSGC as a promising method for stain-free analysis and purification in future cell manufacturing.

As a start, we demonstrated that iSGC can distinguish dead and apoptotic cells from live cells at high accuracy. This is essential in monitoring the population for quality control; checking if the cells under production are healthy (***Campbell et al., 2015***; ***Kolkundkar, 2014***). It is known that using FSC and SSC (or FSC and BSC), dead cells are distinguishable but apoptotic cells are difficult to distinguish from live cells at high accuracy (***Darzynkiewicz et al., 1992***; ***Dive et al., 1992***; ***Shapiro, 2005***; ***Zamai et al., 1993***). In this experiment, we performed a viability and apoptosis analysis of cultured human iPSCs. As shown in ***Figure 3A***, the training label was created using the fluorescence intensity of propidium iodide (PI) and Annexin V, indicators of dead and apoptotic cells, respectively. With these labels, the training data set of pairs of a waveform and label was prepared. To enable effective classification of cells, especially in dead cell discrimination, here, we introduce other stain-free modalities called side scatter GMI (ssGMI) and back scatter GMI (bsGMI) (***Figure 1—figure supplement 1*** and

*Figure 1—figure supplement 2*), which are SSC and BSC variants, respectively, of their fluorescence waveform counterpart in ghost cytometry. Using dGMI and bsGMI modalities for iSGC, it was able to distinguish dead and apoptotic cells from live cells with an AUC of 0.998±0.002 and 0.877±0.007 (*Figure 3D,E* blue solid lines in *Figure 3—figure supplement 1D and E*). In contrast, using the FSC and BSC data with the same labels for SVM-based classification, conventional flow cytometry can only achieve a limited performance of 0.973±0.007 and 0.779±0.011 (gray dashed lines in *Figure 3—figure supplement 1D and E*). Especially, between apoptotic and live cells, both populations overlap in the FSC-BSC scatter plot (*Figure 3C*), and with conventional gating in the FSC-BSC scatter plot (*Hawley and Hawley, 2004*) (region within the red line in *Figure 3B*), apoptotic cells cannot be excluded from live cells. Using iSGC, we can distinguish apoptotic cells from live cells within this gate with an AUC of 0.891±0.012 (*Figure 3F*, blue solid line in *Figure 3—figure supplement 1F*). Therefore, we show that iSGC can classify cells that conventional stain-free scattering intensity-based methods fail to distinguish, proving its significant potentials in monitoring cell viability.

Next, we demonstrated that the iSGC can classify undifferentiated cells from differentiated ones, an important process in the cell production line. This is critical because remaining undifferentiated cells can eventually cause the tumor formation after their transplantation (*Ben-David and Benvenisty,*

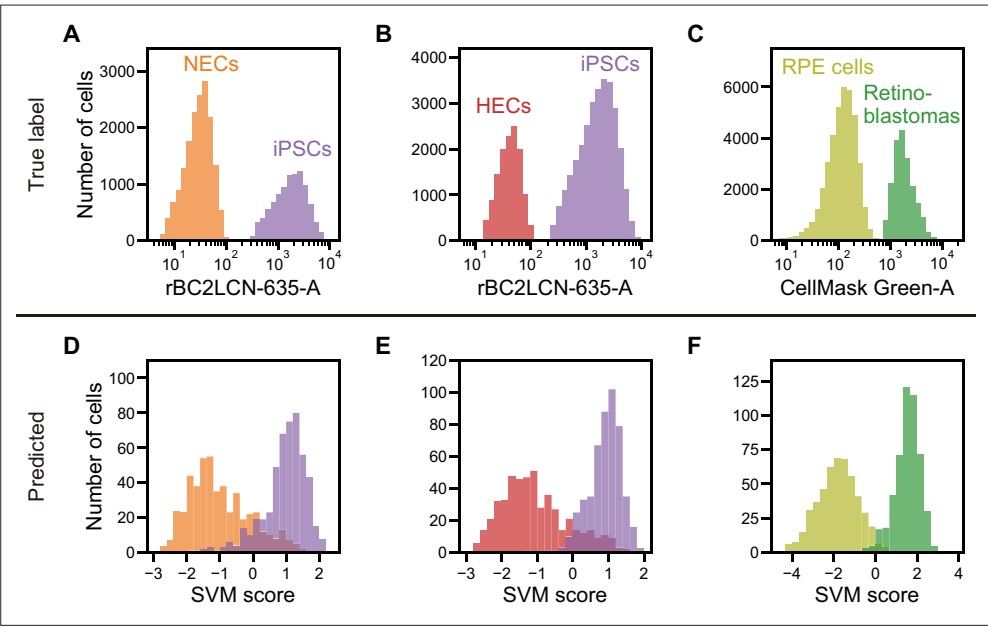

**Figure 4.** Actual fluorescence labels and predicted labels by iSGC for the classification of undifferentiated and cancer cells from iPSC-derived cells. (**A–C**). Histograms of rBC2LCN-635 fluorescence intensity for a mixture of NECs (orange) and iPSCs (violet) (**A**), a mixture of HECs (red) and iPSCs (violet) (**B**), and a histogram of CellMask Green intensity for a mixture of RPE cells (light green) and retinoblastoma (green) (**C**). These markers were used for discriminating cells in a training data set. (**D–F**). SVM score histogram for the iSGC-based classification of NECs and iPSCs (**D**), HECs and iPSCs (**E**), and RPE cells and retinoblastoma (**F**) with dGMI and ssGMI. The colors correspond to the labels in (**A**), (**B**), and (**C**) used for validation. All histograms are the best result of 10 times random sampling. The AUCs for each classification were 0.943 (**D**), 0.951 (**E**), and 0.998 (**F**). The mean and standard deviation of AUCs for the 10 trials in each condition were 0.929±0.008, 0.942±0.007, and 0.994±0.002, respectively (*Figure 4—figure supplement 1*, D, E, and F). dGMI, diffractive ghost motion imaging; HEC, hepatic endodermal cell; iPSC, induced pluripotent stem cell; iSGC, in silico-labeled ghost cytometry; NEC, neuroectodermal cell; RPE, retinal pigment epithelium; ssGMI, side scatter ghost motion imaging; SVM, support vector machine.

The online version of this article includes the following figure supplement(s) for figure 4:

**Figure supplement 1.** Scatter plot gating and ROC curves for the classification of undifferentiated or cancer cells and iPSC-derived cells.

**Figure supplement 2.** Gating for the iSGC analysis in the classification of iPSCs and NECs.

**Figure supplement 3.** Gating for the iSGC analysis in the classification of iPSCs and HECs.

**Figure supplement 4.** Gating for the iSGC analysis in the classification of RPE cells and retinoblastomas.

2011; *Knoepfler, 2009*). Here, we compared undifferentiated human iPSCs with differentiated cells—neuroectodermal cells (NECs) and hepatic endodermal cells (HECs), both derived from the same iPSCs. In the training process, the cells were labeled with an undifferentiation marker which only stains the iPSCs that are undifferentiated. The training label was created by staining the cells with rBC2LCN-635, a fluorescent probe for indicating undifferentiated cells (*Onuma et al., 2013*; *Figure 4A and B* and *Figure 4—figure supplement 1*, A and B). Using dGMI and ssGMI waveforms of each cell type labeled by the undifferentiation marker as training data set, the classifier distinguished iPSCs and NECs with an AUC of 0.929±0.008 (*Figure 4D* and *Figure 4—figure supplement 1D*), and iPSCs and HECs with an AUC of 0.942±0.007 (*Figure 4E* and *Figure 4—figure supplement 1E*), showing its high classification capability. In contrast, using only both FSC and SSC information, the AUCs for the SVM-based classification of each pair of cell types were limited to 0.856±0.009 and 0.697±0.016 (*Figure 4—figure supplement 1*, D and E), respectively.

In addition, we also demonstrated that iSGC can be used to further purify cells of final products by distinguishing contaminating cancer cells. In the final stage of production, the discrimination of contaminants at high accuracy is necessary to remove potentially cancerous cells, which may lead to tumors after their transplantation. We chose retinal pigment epithelium (RPE) cells, the first iPSC-derived cells tested clinically, as a product of regenerative medicine (*Cyranoski, 2017*; *Cyranoski, 2014*; *Garber, 2015*; *Mandai et al., 2017*). We applied our method for classifying a mixed population of RPE cells and Y-79 retinoblastoma cells, model epithelial cancer cells that can potentially remain in the RPE cells (*Reid et al., 1974*). Only the retinoblastoma cells were stained with Cell-Mask Green Plasma Membrane Stain (Invitrogen) prior to mixing the two cell populations. Using dGMI and ssGMI waveforms labeled based on the intensity of CellMask Green as training data set (*Figure 4C* and *Figure 4—figure supplement 1C*), the AUC for classification of RPE cells with Y-79 cells was 0.994±0.002 (*Figure 4F* and *Figure 4—figure supplement 1F*). In contrast, simultaneously using FSC and SSC information for the SVM-based classification, the AUC was limited to 0.966±0.004 (*Figure 4—figure supplement 1F*). This difference becomes substantial when purifying RPE cells at a low false positive rate, or low number of overlooked cancer cells. For instance, to achieve a false positive rate of less than 1%, 92% of the RPE cells can be recovered with iSGC but only 52% of the RPE cells can be recovered with FSC and SSC (see Materials and methods). Therefore, iSGC can be used to distinguish and remove cancerous cells from the final products of cell manufacturing at high sensitivity and specificity compared to conventional stain-free methods.

## White blood cell differentiation

Other than cell manufacturing, iSGC can be used for medical diagnosis based on cell classification. For example, peripheral white blood cell (WBC) differential count is necessary for the diagnosis of diseases such as inflammatory states, sepsis, allergy, and hematologic malignancies (*Blumenreich, 1990*; *Roussel et al., 2012*). The current reference method is a 400 cell count performed manually under a microscope (*Cherian et al., 2010*; *Hubl et al., 1997*; *Roussel et al., 2012*; *Roussel et al., 2010*). However, not only is this labor-intensive and does require skilled technicians, but may also limit accuracy and precision because of the examination of small number of cells and the subjective distinction of cells (*Cherian et al., 2010*; *Hubl et al., 1997*; *Roussel et al., 2012*; *Roussel et al., 2010*). Recently, studies have been ongoing to replace the reference method with flow cytometry (*Cherian et al., 2010*; *Hubl et al., 1997*; *Roussel et al., 2012*; *Roussel et al., 2010*). Still, because one CD marker can differentiate only one cell type at the most, using antibodies for multiple CD markers expands the cost of diagnostic tests to be performed as a routine basis. With iSGC, peripheral WBC differential counting can be performed based on the morphology of the cells without CD markers, but objectively and with thousands of cells.

Here, we demonstrate stain-free peripheral WBC differential classification with iSGC using healthy blood samples spiked with model CD45+ hematopoietic stem cells (HSCs). To train and validate our system, the WBCs were stained with five fluorochrome-conjugated monoclonal antibodies that each bind to one of these types of CD markers; CD45, CD123, CD14, CD16, and CD34. Using these markers, the cells were labeled as neutrophils, lymphocytes, monocytes, eosinophils, basophils, or HSCs (*Figure 5A*), corresponding to a widely performed, five-part peripheral WBC differential (*Hubl et al., 1997*; *Roussel et al., 2010*) and HSC enumeration (*Brocklebank and Sparrow, 2001*; *Venditti et al., 1999*). Here, we additionally utilized two other stain-free modalities which we call forward

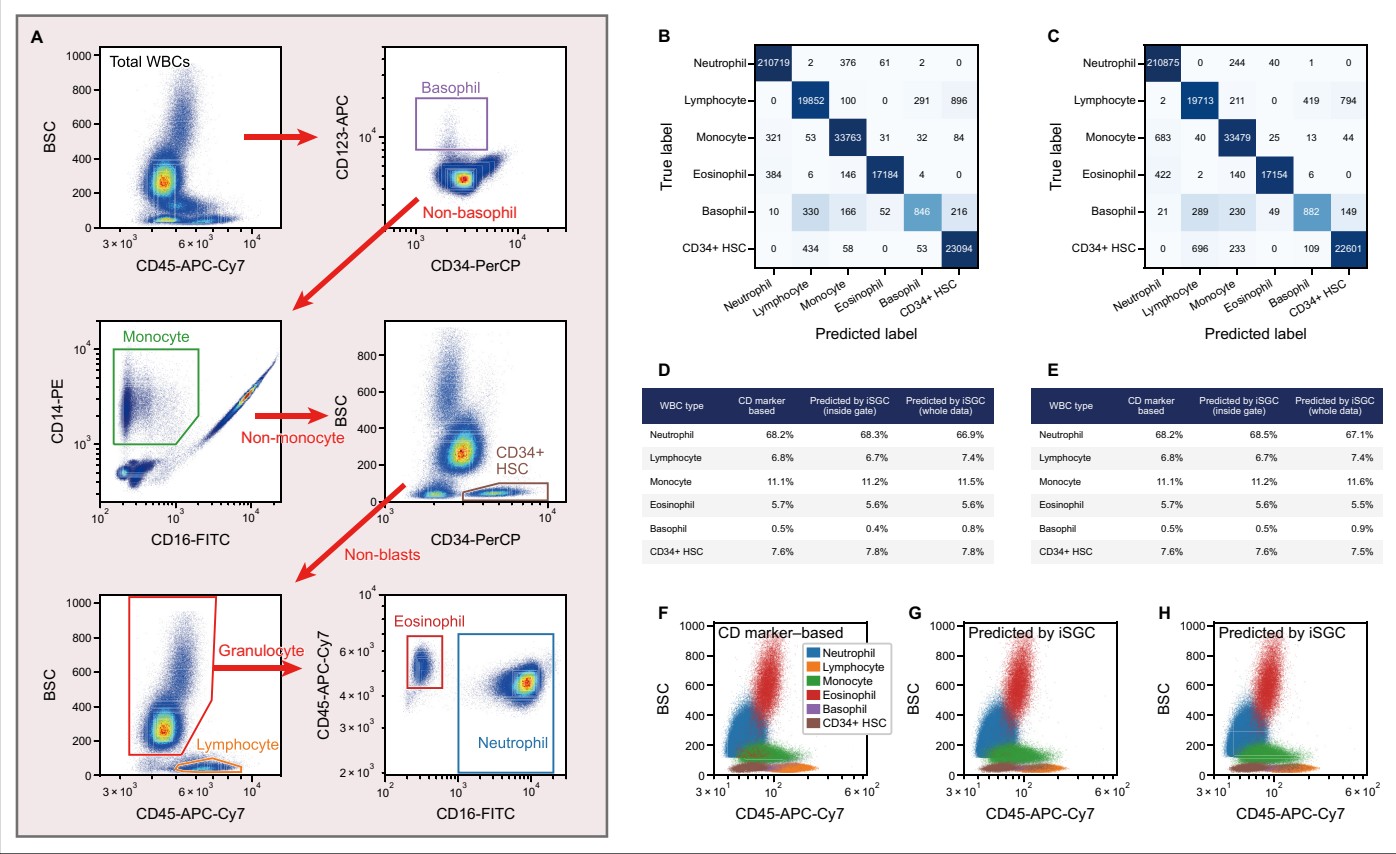

**Figure 5.** Peripheral WBC differential classification with iSGC. (**A**) Flow-cytometry scatter plots used for gating and labeling WBCs. (**B, C**) Confusion matrix for the classification of peripheral WBCs. The training and classification were performed using a sample from the same donor (intra-donor) for (**B**) and samples from different donors (inter-donor) for (**C**). The macro-average F1-scores were 0.911 and 0.904 for (**B**) and (**C**), respectively. (**D, E**) Proportion of each cell type by CD-marker-based labels, iSGC-predicted labels for cells that were gated and given the ground truth labels in (**A**), and iSGC-predicted labels for all cells including those that were not gated in (**A**). Similar to (**B**) and (**C**), the training and classification were performed in an intra-donor manner for (**D**) and an inter-donor manner for (**E**). (**F–H**). Scatter plot of CD45 and BSC for all WBC. The colors in each plot correspond to CD-marker-based labels (**F**), iSGC-predicted labels trained, and predicted in an intra-donor manner (**G**), and iSGC-predicted labels trained and predicted in an inter-donor manner (**H**). The colors in (**G**) and (**H**) correspond to the same labels as (**F**). BSC, back scatter; iSGC, in silico-labeled ghost cytometry; WBC, white blood cell.

The online version of this article includes the following figure supplement(s) for figure 5:

**Figure supplement 1.** Confusion matrix and proportion of cell types for WBC differential classification performed for a different donor sample.

**Figure supplement 2.** Gating for a single run of blood cells obtained from a single donor used for providing the ground truth and performing intra-donor learning for the iSGC analysis in WBC differential classification.

**Figure supplement 3.** Gating for a single run of blood cells obtained from a different donor used for performing inter-donor learning for the iSGC analysis in WBC differential classification.

scatter GMI (fsGMI) and bright field GMI (bfGMI), which are analogs to forward scattering in flow cytometry and bright field images in microscopy, respectively, to improve classification performance (*Figure 1—figure supplement 2*). Also, convolutional neural network (CNN) was adopted to build a classifier for this sorting-free task. After training the iSGC classifier with the CD-marker-based labels, we predicted the peripheral WBCs as one of the cell types using dGMI, bsGMI, fsGMI, bfGMI, FSC, and BSC modalities. When we performed the classification for two donors in an intra-donor manner by training a model using data from one donor and applying it to data from the same donor, macro-average F1-scores of 0.906±0.003 and 0.906±0.002 were obtained, respectively (*Figure 5B* and *Figure 5—figure supplement 1A*). Moreover, when we performed the classification for the same two donors in an inter-donor manner by training a model using a data set from one donor and applying it to a data set from another donor, and vice versa, macro-average F1-scores of 0.901±0.002 and 0.884±0.004 were obtained, respectively (*Figure 5C* and *Figure 5—figure supplement 1B*). Slight

false negatives can be seen for basophils, which can be thought to be due to their limited number in the population used for training and testing the model. In the proportion of each cell in the population (*Figure 5D and E*) and the flow cytometry scatter plot (*Figure 5F,G and H*) compared to the CD-marker-based labeling, the predictions made by iSGC performed well across samples. The class ratio by CD marker and prediction were in good agreement for both intra- and inter-donor predictions, even if the predictions included cells that were not gated and were difficult to clearly label (*Figure 5*). Therefore, we demonstrate that iSGC has the potential to replace CD markers for typical five-part peripheral WBC differential classification.

## Discussion

The advantage of iSGC over conventional label-free flow cytometry modalities with a scalar quantity is that it measures more detailed information of cells for a more accurate cell selection. For instance, with conventional FSC and SSC, each modality yields a single scaler value, and therefore, even combined, the gating is only in a 2D space. However, iSGC utilizes a waveform containing over 100 points and performs classification in this high-dimensional space. While it is difficult for humans to perceive such high-dimensional data, employing machine learning allows us to directly interpret them and perform classification fast enough to enable high-throughput sorting.

Because iSGC utilizes 'image' information, which is a generic indicator of cell morphology just as the 2D cell images, it shares the limitations and advantages with other image-based cell classification methods. For example, iSGC often shows poor classification performances when morphological differences are not easily visually recognizable between cells to be classified. From our experiences, differentiation of CD4 and CD8 T cells are one example which have been challenging to both ghost cytometry and our eyes. On the other hand, transferability of a model can be advantageous: a model developed using different data can be applied to a new sample or application in general, just as those using 2D images can be applied to a new sample or application. Moreover, the iSGC machine is currently equipped with a self-calibration system for controlling the position of flow streams of cells relative to the illumination pattern, which is analogous to a situation where objects come to the same position in the 2D images, consequently enhancing the robustness of models over a long time or between experiments. This is evidenced by the good results of inter-donor classification of the six cell types.

In addition to the modalities of conventional FSC and SSC, dGMI, fsGMI, bsGMI, ssGMI, and bfGMI, we used in this work, the concept of iSGC is not limited to those—other label-free modalities with high dimensions can also be used and combined. For example, imaging techniques such as holography may be able to be adapted for iSGC. What is critical is that such modalities are acquired so that the data can be processed quickly for classification. In the case of holography, the reconstruction of the object image from compressive measurements have been demonstrated (*Brady et al., 2009*; *Lim et al., 2011*; *Marim et al., 2010*), These holographic measurements may be further compressed for classifying the cells based on iSGC.

The potential applications of this method are similar to those of other image-based cell classification, including purification and quality control of cell manufacturing products and diagnostic tests using blood cells, as we demonstrated in this work. These include image-based classification of cell types (*Yoon et al., 2017*), differentiated cells (*Zhang et al., 2018*), cancer cells (*Teramoto et al., 2019*), and WBCs (*Lippeveld et al., 2019*). The advantage over conventional image-based cell classification is that iSGC is able to perform cell classification at higher throughputs which is necessary for analyzing large populations of cells in manufacturing (*Sutermaster and Darling, 2019*) and in diagnostic tests where rare cell detection is often required. Furthermore, in contrast to microscopy image-based cell classification, iSGC-based high-throughput enrichment of the desired cells allows us to use them for their downstream uses in manufacturing as well as molecular assays in further detailed analysis (*Grün and van Oudenaarden, 2015*).

Finally, the concept of iSGC can be used with a variety of ground truth labels. Depending on the characteristics of cell populations and the purpose of classifications, ground truth labels in training data sets can be prepared with various surface markers, genetic reporters (*Zhang et al., 2014*), and functional assays (*Barros et al., 2009*). Furthermore, if we have separated cell populations showing characteristics such as responder versus non-responder cells (*Belzeaux et al., 2012*), or disease versus healthy cells, they can be used as a label training a model to classify and selectively sort the unknown

cells. In addition, it is often the case that molecular labels suitable as ground truth labels are unavailable or less-biased clustering of the morphological information is preferred. One approach for this issue is to use the same high-dimensional modalities for clustering cells based on morphological information via dimension reduction and then adopt these visualized clusters as new labels for training the supervised learning model in iSGC.

In conclusion, we have developed a high-speed, stain-free cytometry that predicts labels based on compressive imaging information without image reconstruction. This staining-free technology will find its applications in a wide range of medical fields such as cell manufacturing in regenerative cell therapy and blood cell enumeration and differentiation in clinical diagnosis. The key to this technology is in turning imaging modalities that are incomprehensible to humans practical for cell characterization by using machine learning to correlate them to biological labels. In the current era where machines can outperform humans, we believe in the potential of utilizing modalities that are machine suitable, rather than human suitable, and that such concept will accelerate not only the field of flow cytometry, but also other areas of science.

## Materials and methods
### Electronics
All photomultiplier tubes (PMTs) used in this work were purchased from Hamamatsu Photonics Inc PMTs of 10 MHz with built-in amplifier (H10723-210 MOD, MOD2, H10723-Y2, A2, MOD2) were used for detecting the dGMI, ssGMI, bsGMI, and fsGMI signals, while PMTs of 200 kHz (H10723-20 Y1, H10723-20 MOD, H10723-210 Y1), 1MHz (H10723-210 MOD3, H10723-20 MOD3), or 10MHz (H10723-210 MOD2) were used to detect fluorescence signals, SSC, and BSC signals. FSC signals were obtained using either a photodetector (PDA100A or PDA100A2, Thorlabs) or a PMT of 200 kHz (H10723-20 MOD, H10723-20-01). Multi-pixel photon counter (MPPC, S13360-6075CS) from Hamamatsu photonics Inc was used to detect the bfGMI. The direct current of dGMI, fsGMI, bfGMI, ssGMI, and FSC signals were cut with an electronic high-pass filter. The PMT signals were recorded with electronic filters using a digitizer (M2i.4932-Exp, Spectrum, Germany) or an FPGA development board (TR4, Terasic) with a homemade analog/digital converter. The digitizer and/or FPGA continually collected a fixed length of signal segments from each color channel at the same time, with a fixed trigger condition applied to the FSC signals.

### Reagents
All reagents were purchased from either FUJIFILM Wako Pure Chemical Corporation, Sigma-Aldrich, or Invitrogen unless otherwise specified. D-PBS (-) (Wako) was used for phosphate-buffered saline (PBS) solution. LIVE/DEAD Fixable Green Dead Cell Stain (Invitrogen) was used by dissolving the solid product in a single vial for 40 assays in 50 µl of dimethyl sulfoxide (DMSO). Pierce 16% Formaldehyde (w/v), Methanol-free (Thermo Fisher Scientific) was used as a fixation reagent.

D-MEM (High Glucose) with L-Glutamine, Phenol Red, and Sodium Pyruvate (Wako) with 10% fetal bovine serum (FBS, Biowest) and 1% Penicillin-Streptomycin (Wako) was used as the HeLa S3 and MIA PaCa-2 cell culture medium. RPMI 1640 medium (Gibco) with 20% FBS (Biowest) and 1% Penicillin-Streptomycin (Wako) was used as the Y-79 cell culture medium. Stem Fit AK02N (Ajinomoto) with all supplements and with 10 µM Y-27632 (Wako) and 0.25 µg/cm² iMatrix-511 silk (Nippi) or iMatrix-511 (Nippi), and 1% Penicillin-Streptomycin (Wako) was used as the iPSC culture medium. Stem Fit AK02N without supplement C and with 10 µM SB431542 (Wako) and 10 µM DMH1 (Wako) was used as the NEC differentiation medium. Stem Fit AK02N without supplement C and with 10 ng/ml Activin A (Wako) and 3 µM CHIR99021 (Wako) was used as endodermal differentiation medium. StemSure DMEM (Wako) with 20% StemSure Serum Replacement (Wako), 1% L-alanyl L-glutamine solution (Nacalai Tesque), 1% Monothioglycerol solution (Wako), 1% MEM non-essential amino acids solution (Nacalai Tesque), and 1% DMSO (Sigma-Aldrich) was used as the HEC differentiation medium. RPMI1640 (FUJIFILM Wako) with 10% FBS (HyClone) and 1% Penicillin-Streptomycin (Wako) was used for thawing the frozen mobilized peripheral blood (MPB) CD34+ stem/progenitor cells.

## Cell preparations

MIA PaCa-2 cells were purchased from RIKEN BRC CELL BANK. HeLa S3 cells, Y-79 retinoblastoma cells were purchased from JCRB Cell Bank. The human iPSCs used were GM25256 iPSCs (*Hayashi et al., 2016*) either obtained from Dr. Bruce Conklin at Gladstone Institutes or purchased from Coriell Institute. Informed consent for the usage of these cells for research was obtained from the cell line donor. RPE cells were provided from HEALIOS K.K. upon permission of use. The usage of iPSCs for the derivation of RPE cells was approved by the ethics committee of HEALIOS. Informed consent for the usage of these iPSCs was obtained by Lonza. MIA PaCa-2 cells, HeLa S3 cells, and both iPSCs were authenticated with short tandem repeat tests and confirmed mycoplasma negative. Y-79 was authenticated and confirmed mycoplasma negative by the supplier. The collection of peripheral blood samples from healthy donors was conducted in accordance with the Declaration of Helsinki and approved by the ethics committee of Juntendo University School of Medicine (IRB#2019091). Mobilized Peripheral Blood CD34+ Stem/Progenitor cells (HSCs) (M34C-1) were purchased from HemaCare. Written informed consent was obtained prior to the collection of samples.

The GM25256 iPSCs were seeded at 2500 cells/cm$^2$ in the iPSC culture medium after each passage. The medium was changed to the same medium without Y-27632 or iMatrix-511 on days 1, 3, and 5. On day 7, the cell samples were collected to be analyzed by flow cytometry.

NECs were differentiated from the above iPSCs with the following procedures. iPSCs were seeded at 2500 cells/cm$^2$ in iPS culture medium (day 0). On day 1, the medium was exchanged to NEC differentiation medium. The medium was changed to the same medium on days 3 and 5. On day 7, the cell samples were collected to be analyzed with iSGC.

HECs were differentiated from the above iPSCs with the following procedures. iPSCs were seeded at 2500 cells/cm$^2$ in iPS culture medium (day 0). On day 1, the medium was exchanged to endoderm differentiation medium. On day 3, the medium was changed to the same medium. On day 5, the medium was exchanged to HEC differentiation medium. On day 7, the cell samples were collected to be analyzed with iSGC.

RPE cells were differentiated from feeder-free hiPSCs established at Lonza, Walkersville according to published protocols (*Baghbaderani et al., 2015*). The hiPSCs were maintained on iMatix-511 (Nippi) with TeSR-E8 medium (StemCell Technologies). To differentiate into neural ectoderm lineage, hiPSCs were cultured with GMEM (Thermo Fisher Scientific) supplemented 20% KnockOut Serum Replacement (Thermo Fisher Scientific), SB431542 and LDN193189 (*Surmacz et al., 2012*). SB431542 and LDN193189 were purchased from FUJIFILM Wako chemical. Then neural ectodermal progenitor cells were differentiated into RPE cells (*Kuroda et al., 2019*). After differentiation, RPE cells were cultured on a laminin-coated 12-well plate. 43.2 μl of 0.5 mg/ml iMatrix-511 (Nippi) was added to 12 ml of PBS, and 1 ml of the solution was added to each well. The well plate was incubated at 37°C under 5% CO2 for 2 hr, and the PBS was removed before use. The thawed RPE cells were seeded onto this laminin-coated well plate and were cultured using an RPE culture medium provided by HEALIOS. The medium was changed every 2 days. The RPE cells were cultured for 8 weeks after thawing before the analysis with iSGC.

## Experimental conditions for iSGC

The cells were flowed through either a quartz flow cell (Hamamatsu) or a polydimethylsiloxane (PDMS)-based microfluidic device using a customized pressure pump and/or a syringe pump (KD Scientific). The quartz flow cell had a channel cross-section dimension of either 250×500, 250×250, or 150×150 μm$^2$ at the measurement position, and when using this, the sheath fluid (IsoFlow, Beckman Coulter) was driven at a pressure of about 25, 85, or 305 kPa, respectively. The PDMS device had a channel with a cross-section dimension of 37×50 μm$^2$ at the measurement position, and when using this, the sheath flow was driven at a pressure of about 185 kPa. The sample fluid was driven at a flow rate between 10 and 40 μl/min.

For the binary classification of the cells, an SVM algorithm using either linear or radial basis function kernel was used. The regularization coefficient and kernel coefficient in the SVM were tuned by grid search. Except for the classification and sorting of HeLa S3 and MIA PaCa-2 cells in *Figure 2*, all trainings and validations of the algorithms were performed using equal amounts of samples for each class label, and the accuracies, receiver operating characteristic (ROC) curves, AUCs, and the macro-average F1-scores were validated with 10 times random sampling and presented with the mean and

standard deviation of the 10 trials. The number of samples was determined as sufficient amount to perform learning and evaluation of the SVM model. SVM score histograms in *Figures 3 and 4* were derived from the trials with the best AUC. The mean and standard deviation of AUCs in the caption of *Figure 3* in the main text were derived from the 10 AUCs, whereas the means and standard deviations of AUCs in *Figure 3—figure supplement 1* and *Figure 4—figure supplement 1* were derived from the mean ROC curve. Therefore, the AUCs will slightly alter. All data and codes are available on Zenodo (*Ugawa et al., 2021*).

For the multiclass classification of six subtypes of WBC, we created a CNN model with two types of feature inputs. For input, we used both waveforms and scalar features derived from GMI (fsGMI, bsGMI, dGMI, and bfGMI) and FSC, BSC features, respectively. bfGMI was obtained by focusing the image plane on the detector. Our model includes five convolutional layers, one fully connected layer, and one softmax layer. Each convolutional layer is followed by batch normalization, rectified linear unit (ReLU), and max pooling layers. Waveforms were used as inputs for the convolutional layer. Scalar features were concatenated to the previous outputs of the fully connected layer. For training, categorical cross-entropy was used as a loss function. The model was trained for 500 epochs with Adam optimizer with a learning rate of 0.00001 and batch size of 1024. If the validation loss did not improve in 30 epochs, we stopped training and adopted the weights with the best validation loss. The macro-average F1-scores were validated with 10-fold stratified splitting of training data and presented with the mean and standard deviation of 10 folds. Confusion matrices and predictive labels for quantifying the in-sample class ratios were derived from the folds with the best macro-average F1-scores. Detailed information such as number of cells from donors are available in the data and codes on Zenodo (*Ugawa et al., 2021*). The number of cells was determined as sufficient amount to perform learning and evaluation of the CNN model.

The throughput of iSGC was deduced with the following procedures. Assuming that cells are evenly spaced with a distance equal to the length of the structured illumination (*Figure 1—figure supplement 1*), resulting in consecutive acquisition of the waveforms, the throughput of iSGC in this ideal condition can be estimated as the inverse of the acquisition time. For instance, if the acquisition time is 100 µs per cell, the throughput can be estimated as 10,000 cells/s. Such estimation of throughput based on acquisition time is often adopted in imaging cytometry (*Diebold et al., 2013*; *Han et al., 2016*; *Han and Lo, 2015*).

## Classification and sorting of HeLa S3 cells and MIA PaCa-2 cells

HeLa S3 and MIA PaCa-2 cells were detached from the culture flask using trypsin. After washing with PBS solution, cells in each suspension were fixed with 1% formaldehyde in PBS for 15 min. The cells were subsequently washed with PBS solution. To the MIA PaCa-2 cells, a combined solution with 4 µl of Fixable Green DMSO solution and 200 µl of 1% Tween 20 (Sigma-Aldrich) in PBS was added. To the HeLa S3 cells, a combined solution with 4 µl of DMSO and 200 µl of 1% Tween 20 in PBS was added. Both cells were incubated for 60 min at room temperature and subsequently washed with PBS solution. Each cell suspension was then mixed to be allowed to flow through the iSGC sorter system comprised of a cell-sorting microfluidic device similar to one previously reported (*Ota et al., 2018*). The SVM used for sorting was trained with 300 cells of each cell type. The training data was created with consecutive waveforms obtained from the mixed sample.

A portion of this mixed cell suspension was used to measure the FSC, SSC, and fluorescence intensity of the presort mixture by using JSAN (Bay bioscience) to obtain *Figure 2A and B* and dashed line of *Figure 2—figure supplement 1B*. Total of 6000 events were measured, from which 5558 cells were gated with FSC-SSC. Within this population, a threshold was drawn for the green fluorescence intensity to obtain 3350 positive cells and 2208 negative cells. After collecting the sorted suspension from the iSGC sorter, the FSC, SSC, and green fluorescence intensity were measured using JSAN to confirm the purity and obtain *Figure 2E*. Total of 6000 events were measured, from which 5193 cells were gated with FSC-SSC. Within this population, a threshold was drawn for the green fluorescence intensity to obtain 5054 positive cells and 139 negative cells. The SVM scores in *Figure 2C*, the blue solid line in *Figure 2D*, and the confusion matrix in *Figure 2—figure supplement 1C* were reproduced from the saved SVM parameters of the FPGA and the obtained waveform during sorting. Full gating procedures are available in *Figure 2—figure supplement 2*.

## Classification of live, dead, and apoptotic iPSCs

For dissociating iPSCs, the culture medium for the iPSCs was changed to 0.5 mM ethylenedi-aminetetraacetic acid (EDTA) in PBS solution and incubated for 3–5 min. Then medium was changed back to iPSC culture medium, and the cells were detached using a cell scraper. After washing with binding buffer, the cells were stained with PI (Medical & Biological Laboratories) and Annexin V-FITC (Medical & Biological Laboratories) for 15 min and then allowed to flow through the iSGC system without washing. The cells were first gated using the FSC-BSC scatter plot to remove debris (*Figure 3—figure supplement 1A*) and then gated using the FSC height versus width scatter plot to remove doublets. The gated cells were further gated and labeled as live, dead, or apoptotic cells (*Figure 3A*) according to the fluorescence intensity of PI and Annexin V-FITC. Within this data, 1000 cells selected randomly (without overlap) with equal number of cells for each label were used as training and testing data. Full gating procedures are available in *Figure 3—figure supplement 2*.

## Classification of undifferentiated and differentiated cells

For dissociating iPSCs, NECs, and HECs, the culture medium for these cells was changed to 0.5 mM EDTA in PBS solution and incubated for 3 min. Then, the cells were detached from the culture flask using a cell scraper. After washing with buffer solution ( 0.1% FBS, 0.5 mM EDTA in D-PBS), each was stained with Calcein AM (Dojindo) and rBC2LCN-635 (Wako), both at a concentration of 1 µg/ml, for 60 min on ice and then washed with buffer solution. For the classification of undifferentiated cells (iPSCs) and differentiated cells (NECs or HECs), each of the cells was mixed at equal concentration. The mixed suspension was allowed to flow through the iSGC system. The cells were first gated using FSC-SSC scatter plot to remove dead cells and debris, then gated using the FSC height versus width scatter plot to remove doublets, and again gated using FSC-Calcein AM scatter plot to remove remaining dead cells. The gated cells were further gated and labeled as undifferentiated or differentiated cells according to the fluorescence intensity of rBC2LCN-635 and Calcein AM (*Figure 4—figure supplement 1A and B*). Within this data, 5000 and 1000 cells selected randomly (without overlap) with equal number of undifferentiated and differentiated cells were used as training and testing data, respectively. Full gating procedures are available in *Figure 4—figure supplement 2* and *Figure 4—figure supplement 3*.

## Classification of RPE and retinoblastoma cells

RPE cells were detached from the well plate using trypsin (provided by HEALIOS). Y-79 retinoblastoma cells were taken out from the culture flask by pipetting. After washing each cell with PBS solution, 1 ml of PBS with 1 µl of CellMask Green Plasma Membrane Stain (Invitrogen) was added to the pellet of Y-79 cells and was incubated at  37°C for 15 min. After staining, the cells were washed with PBS solution and PI was added. The RPE cells and Y-79 cells were mixed before being allowed to flow through the iSGC system. The cells were first gated using FSC-SSC scatter plot to remove dead cells and debris, then gated using the FSC height versus width scatter plot to remove doublets. The gated cells were further gated and labeled as RPE cells or retinoblastomas according to the fluorescence intensity of CellMask Green and PI (*Figure 4—figure supplement 1C*). Within this data, 5000 and 1000 cells selected randomly (without overlap) with equal numbers of RPE cells and retinoblastomas were used as training and testing data, respectively. Full gating procedures are available in *Figure 4—figure supplement 4*.

The recovery rate, which is equivalent to the true positive rate, of RPE cells was determined from the mean ROC curve obtained from 10 times random sampling (*Figure 4—figure supplement 1C*). At a false positive rate of 0.01, the true positive rate was 0.92 and 0.52 for the classification with iSGC and with FSC/SSC, respectively. In actual scenarios, the number of cancer cells will be substantially fewer than the number of RPE cells. Still, the true positive rates and false positive rates will be equal to the case where the numbers of cells are equal.

## Differential classification of WBCs

Fresh peripheral blood samples from two healthy volunteers were drawn into tubes containing EDTA. The samples were stained by adding fluorochrome-conjugated monoclonal antibodies and protecting them from light for 15 min at room temperature. The antibodies used were: FITC anti-human CD16 (302006, Lot no. B319858, BioLegend), PE anti-human CD14 (367104, Lot no. B274117, BioLegend),

APC/Cy7 anti-human CD45 (304014, Lot no. B325539, BioLegend), added at 40× dilution and APC anti-human CD123 (306012, Lot no. B309938, BioLegend) added at 20× dilution. The frozen mobilized peripheral blood CD34$^+$ hematopoietic stem/progenitor cells (HSCs) were thawed by immediately placing them into a 37°C water bath and transferring them into a pre-warmed growth medium (RPMI1640 with 10% FBS and 1% Penicillin-Streptomycin). Then the cells were stained by adding fluorochrome-conjugated monoclonal antibodies and protecting them from light for 30 min at room temperature. The antibodies used here were APC/Cy7 anti-human CD45 (304014, Lot no. B325539, BioLegend) added at 13× dilution and PerCP anti-human CD34 (343520, Lot no. B266937, BioLegend) added at 10× dilution. Next, all three samples were lysed for 10 min with a freshly prepared working solution of Lysing Solution (349202, BD Bioscience). After removing the supernatant, the samples were washed with PBS containing 2% bovine serum albumin (BSA).

Each healthy blood sample was spiked with the CD34$^+$ HSCs so that the ratio of HSCs was less than 10%. The final concentration of each spiked sample was adjusted to about $5 \times 10^5$ cells/ml in PBS containing 2% BSA and allowed to flow through the iSGC analyzer system. The total events were gated using FSC-BSC scatter plot to remove dead cells and debris, then gated using the FSC height versus width scatter plot to remove doublets. WBC subtypes were subsequently gated in the order of basophils, monocytes, HSCs, lymphocytes, neutrophils, and eosinophils based on well-established gating schemes (*Fujimoto et al., 2000*; *Han et al., 2008*; *Hubl et al., 1996*; *Roussel et al., 2012*; *Roussel et al., 2010*; *Venditti et al., 1999*). For validation, a portion of each spiked sample was measured with JSAN (Bay bioscience). After gating the cells, each set of cell types were split into 70% training data and 30% test data. For training the model, 10-fold stratified cross validation was used. Classification was performed on the whole test data. Full gating procedures are available in *Figure 5—figure supplement 2* and *Figure 5—figure supplement 3*.

## Acknowledgements

The authors acknowledge HEALIOS KK for providing RPE cells. The authors thank the Nano-Processing Facility, National Institute of Advanced Industrial Science and Technology, Japan for the fabrication of DOE. This work was supported by JST-PRESTO, Japan, Grant numbers JPMJPR14F5 to SO, JPMJPR17PB to RH, and JPMJPR1302 to IS and partially supported by funds of a visionary research program from Takeda Science Foundation, the Mochida Memorial Foundation for Medical and Pharmaceutical Research, and the Nakatani Foundation for Advancement of Measuring Technologies in Biomedical Engineering. The work is based on results obtained from a project commissioned by the New Energy and Industrial Technology Development Organization (NEDO).

## Additional information

### Competing interests

Masashi Ugawa: Former employee and holds shares of stock options of ThinkCyte, Inc. Has filed patent applications related to in silico-labeled ghost cytometry method. Patent number PCT/US2019/36849. Yoko Kawamura: Employee and holds share of stock options of ThinkCyte, Inc. Has filed patent applications related to in silico-labeled ghost cytometry method. Patent numbers PCT/JP2016/082089, PCT/US2019/36849. Keisuke Toda: Employee and holds share of stock options of ThinkCyte, Inc. Has filed patent applications related to in silico-labeled ghost cytometry method. Patent number PCT/JP2021/013478. Kazuki Teranishi, Hiroaki Adachi, Keiji Nakagawa: Employee and holds share of stock options of ThinkCyte, Inc. Hikari Morita: Employee and holds shares of stock options of ThinkCyte, Inc. Ryo Tamoto, Hiroko Nomaru: Employee of ThinkCyte. Keiki Sugimoto: Former employee and holds share of stock options of ThinkCyte, Inc. Yuri An: Employee of ThinkCyte, Inc. Yusuke Konishi, Seiichiro Tabata: Employee of Sysmex Corp. Yohei Hayashi, Hiroyuki Noji: Holds shares of stock options of ThinkCyte, Inc. Issei Sato: Founder and shareholder of ThinkCyte, Inc. Has filed patent applications related to the in silico-labeled ghost cytometry method. Patent numbers PCT/JP2016/082089, PCT/US2019/36849. Ryoichi Horisaki: Founder and shareholder of ThinkCyte, Inc. Has filed patent applications related to the in silico-labeled ghost cytometry method. Patent numbers PCT/JP2016/055412, PCT/JP2016/082089, PCT/JP2018/005237, PCT/US2019/36849. Sadao Ota:

Founder and shareholder of ThinkCyte, Inc. Has filed patent applications related to the in silico-labeled ghost cytometry method. Patent numbers PCT/JP2016/055412, PCT/JP2016/082089, PCT/JP2018/005237, PCT/US2019/36849, PCT/JP2021/013564, PCT/JP2021/013478. The other authors declare that no competing interests exist.

## Funding

| Funder | Grant reference number | Author |
| --- | --- | --- |
| Takeda Science Foundation | | Sadao Ota |
| New Energy and Industrial Technology Development Organization | | Keiji Nakagawa |
| Japan Science and Technology Agency | JPMJPR14F5 | Sadao Ota |
| Mochida Memorial Foundation for Medical and Pharmaceutical Research | | Sadao Ota |
| Nakatani Foundation for Advancement of Measuring Technologies in Biomedical Engineering | | Sadao Ota |
| Japan Science and Technology Agency | JPMJPR1302 | Issei Sato |
| Japan Science and Technology Agency | JPMJPR17PB | Ryoichi Horisaki |

The funders had no role in study design, data collection and interpretation, or the decision to submit the work for publication.

## Author contributions

Masashi Ugawa, Conceptualization, Data curation, Formal analysis, Investigation, Methodology, Software, Visualization, Writing – original draft, Writing – review and editing, Developed optical setups; Yoko Kawamura, Investigation, Methodology, Writing – original draft, Resources, Writing – review and editing, Developed microfluidic devices, Performed experiments; Keisuke Toda, Investigation, Methodology, Developed optical setups; Kazuki Teranishi, Investigation, Resources; Hikari Morita, Hiroaki Adachi, Ryo Tamoto, Formal analysis, Software; Hiroko Nomaru, Keiki Sugimoto, Investigation, Methodology, Resources; Keiji Nakagawa, Funding acquisition, Investigation, Methodology, Resources; Evgeniia Borisova, Methodology, Resources; Yuri An, Seiichiro Tabata, Misa Imai, Resources, Supervision; Yusuke Konishi, Methodology, Supervision; Soji Morishita, Tomoiku Takaku, Supervision; Marito Araki, Methodology, Resources, Supervision; Norio Komatsu, Issei Sato, Software, Supervision; Yohei Hayashi, Conceptualization, Methodology, Resources, Supervision; Ryoichi Horisaki, Conceptualization, Funding acquisition, Project administration, Supervision, Writing – original draft, Writing – review and editing; Hiroyuki Noji, Formal analysis, Software, Supervision; Sadao Ota, Conceptualization, Funding acquisition, Investigation, Project administration, Supervision, Writing – original draft, Writing – review and editing

## Author ORCIDs

Masashi Ugawa http://orcid.org/0000-0002-7196-5737
Ryo Tamoto http://orcid.org/0000-0003-0626-1888
Yohei Hayashi http://orcid.org/0000-0001-5490-7052
Sadao Ota http://orcid.org/0000-0001-6004-488X

## Ethics

Human subjects: The collection of peripheral blood samples from healthy individuals was conducted in accordance with the Declaration of Helsinki, and approved by the ethics committee of Juntendo University School of Medicine (IRB#2019091).

Decision letter and Author response
Decision letter https://doi.org/10.7554/eLife.67660.sa1
Author response https://doi.org/10.7554/eLife.67660.sa2

## Additional files

### Supplementary files
• Transparent reporting form

### Data availability
All original measurement data and codes for analysis are deposited in Zenodo (doi:https://doi.org/10.5281/zenodo.5656641).

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
