## [Editor Report]

This paper explores a novel approve to sorting cells without the use of fluorescent labeling using a light diffraction method called ghost cytometry. This paper first demonstrates this capability with commercial cell lines and then sorting hematopoietic cells from a patient sample.

---

## [Decision Letter]

**Decision letter after peer review:**

Thank you for submitting your article "In silico-labeled ghost cytometry" for consideration by *eLife*. Your article has been reviewed by 2 peer reviewers, one of whom is a member of our Board of Reviewing Editors, and the evaluation has been overseen by Mone Zaidi as the Senior Editor. The following individual involved in review of your submission has agreed to reveal their identity: Gregory R. Johnson (Reviewer #2).

Essential revisions:

1) Address all the comments from the two reviewers; there are concerns regarding applicability and limitations brought up by both reviewers.

2) Reviewer two noted that the presentation could be improved to enhance readability.

*Reviewer #1:*

This novel technique provides a method of cell sorting without the conventional use of fluorescent probes which can effect cell viability. They also showed that ghost cytometry can differentiate between live and apoptotic cells giving a more accurate reflection of cell viability. Retaining specimen viability is particularly important for obtaining cell lines and for therapeutic applications such BM transplant or CAR T therapy.

This paper also demonstrates the ability to generate an accurate white cell differential which could have diagnostic utility to potentially replace manual counts which has interobserver biases. However, this paper did not demonstrate the capacity to differentiate B and T cells, count blasts or detect pathologic hematopoietic cells. It would be interesting to see if this platform could identify cells of different levels maturity as hematopoietic pathologies lie all along the maturation spectrum.

While separating cells at different levels of maturation was demonstrated in cell lines, this is differentiating undifferentiated cells from terminally differentiated cells and does not demonstrate intermediate stages of differentiation.

CD34 should probably have been added to the flow panel to demonstrate if ghost cytometry can count blasts. A white cell differential is incomplete without this capacity.

Another marker that would have been helpful would be using CD138 for plasma cells. Plasma cells are known to be inaccurately counted by flow cytometry. Demonstrating an accurate plasma cell count on ghost cytometry compared to a manual differential could have provided a potential area of superiority diagnostically of ghost cytometry compared to flow cytometry.

The white cell differential was also only done on one patient. Having several patient samples should have been used to validate this method.

*Reviewer #2:*

Flow cytometry typically relies on images and/or multiple chemical labels to identify cellular phenotypes. By training a machine-learning based model on a ground-truth labeled dataset, Ugawa et al. demonstrate that some cellular phenotypes can be accurately determined from a one-dimensional waveform of a cell passing through a field of structured illumination without the use of chemical labels. In combination with ultra-fast cell sorting this method opens the door for tasks where sorted cells may be utilized for downstream applications where staining is undesirable. This manuscript adds to growing number of applications of "in silico labeling" where machine-learning models can be utilized to predict chemical labels from unlabeled samples.

Overall, the approach is well described and the results are promising. The authors evaluate their methods with diverse tasks, but the evaluation procedure may not reflect real-world deployment of such a method.

The manuscript demonstrates the performance of several cell-phenotype classification tasks. After sample preparation, the cells are stained and cell-phenotype labels are assigned according to gates based on ground-truth stain read-out, whereas cells falling outside of the gates are discarded. The labels of the gated cells, in conjunction with their GMI signals are used as training and test data for the machine learning models, and the test results are reported. "Discarded" cells based on the ground-truth stains are not represented in the classification results. If a model were trained and applied to a new sample, those "discarded" populations will be presented to the classifier. The evaluation presented here likely would not reflect the performance of the model relative to a new sample.

As described in the methods, the training and test data are derived from the same sample. If such a model were to be implemented, intra-experiment variation may play a significant role in the accuracy of these results. It is therefore difficult to determine how this method would function when deployed in other settings.

I feel that several points may be addressed to significantly improve the manuscript:

The inclusion of a "discarded" label in the classification results, or otherwise address the "discarded" population problem. Without such, it is very difficult to evaluate the results presented here.

Evaluation of whether the model would generalize well to new samples.

It is not clear what the limitations of this method are. Are there applications where iSGC falls flat? Can we apply multiple models from different data to a new application?

Overall the discussion of flow versus image cytometry (starting at line 60) could be improved. Some claims are without reference, and discussions about the time scales of signal analysis in flow and image cytometry would be useful for more general audiences.

Some of the claims about future applications of these methods seem very strong. It would be useful to provide perspectives from other manuscripts that support discussion of future applications.

There are some compression artifacts in the scatter plot figures that make them difficult to read.

It may be worth considering a more colorblind friendly or perceptually-uniform color map for figures as well.

The "b" in the description of figure 5 should be a "(B)"

---

## [Author Response]

Reviewer #1:This novel technique provides a method of cell sorting without the conventional use of fluorescent probes which can effect cell viability. They also showed that ghost cytometry can differentiate between live and apoptotic cells giving a more accurate reflection of cell viability. Retaining specimen viability is particularly important for obtaining cell lines and for therapeutic applications such BM transplant or CAR T therapy.This paper also demonstrates the ability to generate an accurate white cell differential which could have diagnostic utility to potentially replace manual counts which has interobserver biases. However, this paper did not demonstrate the capacity to differentiate B and T cells, count blasts or detect pathologic hematopoietic cells. It would be interesting to see if this platform could identify cells of different levels maturity as hematopoietic pathologies lie all along the maturation spectrum.While separating cells at different levels of maturation was demonstrated in cell lines, this is differentiating undifferentiated cells from terminally differentiated cells and does not demonstrate intermediate stages of differentiation.

We first would like to thank the reviewer 1 for her or his appreciation on the particular importance of our methods named in silico ghost cytometry (iSGC) in the field of cell therapy as well as its potential utility in a wide range of diagnosis. As the reviewer commented, differentiation of further various types of cells including B and T cells, pathological hematopoietic cells, and cells at intermediate stages of differentiation would be of great interest, which motivate us to explore the exciting future applications of our iSGC by classifying more detailed morphological information.

CD34 should probably have been added to the flow panel to demonstrate if ghost cytometry can count blasts. A white cell differential is incomplete without this capacity.Another marker that would have been helpful would be using CD138 for plasma cells. Plasma cells are known to be inaccurately counted by flow cytometry. Demonstrating an accurate plasma cell count on ghost cytometry compared to a manual differential could have provided a potential area of superiority diagnostically of ghost cytometry compared to flow cytometry.

We first appreciate Reviewer 1’s critical comments on iSGC’s demonstration for the classification of peripheral white blood cells (WBC). Following the recommendation, we performed additional experiments where iSGC was used to differentiate the five normal WBC types plus spiked CD34 positive hematopoietic stem/progenitor cells, resulting in demonstration of accurate differentials of these six cell types (Figure 5 and Figure S7). The results are shown as confusion matrices in Figure 5B and 5C as well as the ratios of each cell type in Figure 5D and 5E in the modified manuscript. When we performed intra-donor classifications for two donors (healthy donors), macro-average F1-scores of 0.906 ± 0.003 and 0.906 ±0.002 were obtained, respectively. Here the intra-donor classifications were performed by training a model using a data set from one donor and applying it to a data set for the same donor. When we performed inter-donor classifications for the same two donors, macro-average F1-scores of 0.884 ± 0.004 and 0.901 ± 0.002 were obtained, respectively. Here the inter-donor classifications were performed by training a model using a data set from one donor and applying it to a data set from another donor, and vice versa. We added explanations about these newly added experiments and their results in the main text in page 7 as well as in the method section.

Regarding the suggestion of using CD138 cells for plasma cells, we agree that it is of interest and expands the utility of label-free ghost cytometry if it becomes possible to predict cell types even if suitable molecular markers are not available. However, it is out of scope of the concept of this work, *in silico* labeling, which is an approach for predicting molecular staining/markers from label-free morphological information [1]. In other words, the molecular labels in conventional flow cytometry are used as ground truth labels in ghost cytometry, but they are not compared to claim superiority of ghost cytometry.

Nevertheless, we appreciate the reviewer’s sharp and constructive comment. Predicting cell types of which molecular staining methods suitable as ground truth is unavailable is quite interesting and potentially powerful. We added sentences explaining its future importance and potentials in the Discussion section in page 10 of the manuscript as following:

“In addition, it is often the case that molecular labels suitable as ground truth labels are unavailable or less-biased clustering of the morphological information is preferred. One approach for this issue is to use the same high-dimensional modalities for clustering cells based on morphological information via dimension reduction and then adopt these visualized clusters as new labels for training the supervised learning model in iSGC.”

[1] Christiansen, E. M., Yang, S. J., Ando, D. M., Javaherian, A., Skibinski, G., Lipnick, S., Mount, E., O’Neil, A., Shah, K., Lee, A. K., Goyal, P., Fedus, W., Poplin, R., Esteva, A., Berndl, M., Rubin, L. L., Nelson, P., and Finkbeiner, S. (2018). In Silico Labeling: Predicting Fluorescent Labels in Unlabeled Images. Cell, *173*(3), 792–803.e19.

The white cell differential was also only done on one patient. Having several patient samples should have been used to validate this method.

We addressed this comment when we made a response to the 1st comment: the six cell type differentials were performed on two donors in both intra-donor and inter-donor manners. The results were consistent in both manners as shown in the modified Figure 5 as well as good macro-average F1-scores: it was recorded as 0.906 ± 0.003 when a model from donor 1 was applied to donor 1, as 0.906 ±0.002 when a model from donor 2 was applied to donor 2, and as 0.884 ± 0.004 and 0.901 ± 0.002 between donors when a model from donor 1 was applied to donor 2 and vice versa, respectively.

Reviewer #2:Flow cytometry typically relies on images and/or multiple chemical labels to identify cellular phenotypes. By training a machine-learning based model on a ground-truth labeled dataset, Ugawa et al. demonstrate that some cellular phenotypes can be accurately determined from a one-dimensional waveform of a cell passing through a field of structured illumination without the use of chemical labels. In combination with ultra-fast cell sorting this method opens the door for tasks where sorted cells may be utilized for downstream applications where staining is undesirable. This manuscript adds to growing number of applications of "in silico labeling" where machine-learning models can be utilized to predict chemical labels from unlabeled samples.Overall, the approach is well described and the results are promising. The authors evaluate their methods with diverse tasks, but the evaluation procedure may not reflect real-world deployment of such a method.The manuscript demonstrates the performance of several cell-phenotype classification tasks. After sample preparation, the cells are stained and cell-phenotype labels are assigned according to gates based on ground-truth stain read-out, whereas cells falling outside of the gates are discarded. The labels of the gated cells, in conjunction with their GMI signals are used as training and test data for the machine learning models, and the test results are reported. "Discarded" cells based on the ground-truth stains are not represented in the classification results. If a model were trained and applied to a new sample, those "discarded" populations will be presented to the classifier. The evaluation presented here likely would not reflect the performance of the model relative to a new sample.As described in the methods, the training and test data are derived from the same sample. If such a model were to be implemented, intra-experiment variation may play a significant role in the accuracy of these results. It is therefore difficult to determine how this method would function when deployed in other settings.

We first would like to thank the reviewer 2 for clear understanding and importance of our technology as “this method opens the door for tasks where sorted cells may be utilized for downstream applications where staining is undesirable. This manuscript adds to growing number of applications of “in silico labeling” where machine-learning models can be utilized to predict chemical labels from unlabeled samples.” Regarding the critical comments on the “discarded” populations and inter-experimental variability, we modified the manuscript to show that iSGC analysis is robustly consistent even when the “discarded” populations are included. Moreover, we demonstrated that inter-donors samples can be robustly classified using an implemented model. These are described in more detail in the response to the “Recommendations for the authors”. Overall, sharp comments from the reviewer 2 did help us to significantly improve the detailed presentation of our work. We really appreciate these comments again.

I feel that several points may be addressed to significantly improve the manuscript:The inclusion of a "discarded" label in the classification results, or otherwise address the "discarded" population problem. Without such, it is very difficult to evaluate the results presented here.

We appreciate the reviewer’s comment as we could make the manuscript more substantial and solid by addressing it. While it is not straightforward to compare the ratio of cell types defined using CD markers and that of those including cells which are excluded using CD markers, we agree its importance from the practical perspective.** **

We first addressed this question by including the “discarded” labels (after single gating) in the results of predicting ratios of each cell type when we performed new experiments of six-class-white blood cell (WBC) differentials, as shown in Figure 5D and 5E of the modified manuscript. As a result, the cells defined using CD markers account for the majority (95.6% and 94.4%) of the whole singlet cell population such that the inclusion of a “discarded” label did not affect the classification results of the six class WBC differentials significantly.

Moreover, as shown in Figure 5E, even when a model trained using a cell sample from one donor is applied to a new sample from another donor, the performance of the model was maintained high. In addition, this model maintained its performance even when the “discarded” labels in a new sample were included.

Evaluation of whether the model would generalize well to new samples.

We appreciate this critical comment. In the modified manuscript, we newly performed classification of six (five normal white blood cells + spiked CD34 positive hematopoietic stem/progenitor cells) cell types for two donors in both intra-donor and inter-donor manners (Figure 5 and Figure S7). The intra-donor classifications were performed by training a model using a data set from one donor and applying it to a data set for the same donor. The inter-donor classifications were performed by training a model using a data set from one donor and applying it to a data set from another donor, and vice versa.

As a result, consistently good performances are shown as confusion matrices in Figure 5B and 5C as well as the ratios of each cell type in Figure 5D and 5E in the modified manuscript. When we performed the intra-donor classifications for two donors, macro-average F1-scores of 0.906 ± 0.003 and 0.906 ±0.002 were obtained, respectively. When we performed the inter-donor classifications for the same two donors, macro-average F1-scores of 0.884 ± 0.004 and 0.901 ± 0.002 were obtained, respectively. We added explanations about these newly added experiments and their results in the main text in page 7 as well as in the method section.

It is not clear what the limitations of this method are. Are there applications where iSGC falls flat? Can we apply multiple models from different data to a new application?

We thank the reviewer for constructive comments. As a response to the 1st question, simplest speaking, iSGC can fall flat when morphological differences are not easily visually recognizable between cells to be classified, which is often the case in reality. Differentiation of CD4 and CD8 T cells are one example which have been challenging to both ghost cytometry and our eyes.

Regarding the second question, while we have not ever tried inter-application trials, multiple models from different data can be applied to a new application in principle, just as those from different 2D images can be applied to a new application in general. This is because the temporal waveforms in iSGC are generic indicators of cell morphology just as the 2D images are. Moreover, the iSGC machine is currently equipped with a self-calibration system for controlling the position of flow streams of cells relative to the illumination pattern, which is analogous to a situation where objects come to the same position in the 2D images, consequently enhancing the robustness of models over a long time or between experiments. This is evidenced with the results of inter-donor classification of the six cell types.

These discussions are additionally included in the main text of the manuscript in page 8 as following:

“Because iSGC utilizes “image” information, which are generic indicators of cell morphology just as the 2D cell images, it shares the limitations and advantages with other image-based cell classification methods. […] This is evidenced with the good results of inter-donor classification of the six cell types. ”

Overall the discussion of flow versus image cytometry (starting at line 60) could be improved. Some claims are without reference, and discussions about the time scales of signal analysis in flow and image cytometry would be useful for more general audiences.

Thank you for the suggestion which helps us improve the readability of this paper. In the discussion of flow versus image cytometry starting from line 60 in page 2 of the manuscript, we added references and modifications as following:

“On the other hand, microscopic image-based cell classification of unstained cells is free from such limitations of molecular labeling and is a promising approach for evaluating cell functions or potentials in fields such as cell manufacturing (Buggenthin et al., 2017; Chang et al., 2017; Niioka et al., 2018). […] In contrast, conventional flow-based cell sorting systems that process simple cell information such as total fluorescence intensity fast enough to operate at around 10,000 cells/s (Sutermaster and Darling, 2019).”

Some of the claims about future applications of these methods seem very strong. It would be useful to provide perspectives from other manuscripts that support discussion of future applications.

Thank you for the suggestion to add discussions that refer to other works that relate to the mentioned applications.

Regarding the future extension of iSGC technology, we added references and modified the relevant sentences in page 8-9 of the manuscript as following:

“In addition to the modalities of conventional FSC and SSC, dGMI, fsGMI, bsGMI, ssGMI, and bfGMI we used in this work, the concept of iSGC is not limited to those – other label-free modalities with high dimensions can also be used and combined. […] Even though perceptually it may look difficult to extract features, machine learning algorithms can extract features from the raw image [Sinha et al. Optica 2017, Rivenson et al. Light:Science and Applications 2018], because after all, the final image is derived from the raw image.”

Regarding the biological applications of evaluating cells based on image information, we added references and modified the relevant sentences in page 8 of the manuscript as following:

“The potential applications of this method are similar to those of other image-based cell classification, including purification and quality control of cell manufacturing products and diagnostic tests using blood cells, as we demonstrated in this work. […] Furthermore, in contrast to microscopy image-based cell classification, iSGC-based high-throughput enrichment of the desired cells allows us to use the desired cells for their downstream uses in manufacturing as well as molecular assays in further detailed analysis (Grün and Van Oudenaarden, 2015).”

There are some compression artifacts in the scatter plot figures that make them difficult to read.It may be worth considering a more colorblind friendly or perceptually-uniform color map for figures as well.The "b" in the description of figure 5 should be a "(B)"

We would like to thank the reviewer for her or his careful reading of our paper. Regarding the first point “some compression artifacts in the scatter plot figures ” we could not tell which figure might have the compression artifacts. So we would appreciate it if we could tell which figure to modify concretely. For other points, we have revised the typos and color-map of figures which the reviewer pointed out